# Evolution of research funding for neglected tropical diseases in Brazil, 2004–2020

**Gabriela Bardelini Tavares Melo**[1,2]*, **Antonia Angulo-Tuesta**[1], **Everton Nunes da Silva**[1], **Thaís da Silva Santos**[1], **Liza Yurie Teruya Uchimura**[3], **Marcos Takashi Obara**[1]

**1** Graduate Program in Health Sciences and Technologies, Faculty of Ceilândia - University of Brasilia, Federal District, Brasilia, Brazil, **2** Department of Science and Technology, Ministry of Health, Federal District, Brasilia, Brazil, **3** Hospital do Coração, São Paulo State, São Paulo, Brazil

* gabrielabtm@gmail.com

**Data Availability Statement:** The database is available at https://osf.io/mwgrp/?view_only=9a28993d8afa47e08c286caeb8c1e03f and https://pesquisasaude.saude.gov.br/.

## Abstract

Neglected tropical diseases are a global public health problem. Although Brazil is largely responsible for their occurrence in Latin America, research funding on the subject does not meet the population's health needs. The present study analyzed the evolution of research funding for neglected tropical diseases by the Ministry of Health and its partners in Brazil, from 2004 to 2020. This is a retrospective study of data from investigations registered on Health Research (*Pesquisa Saúde* in Portuguese), a public repository for research funded by the Ministry of Health's Department of Science and Technology. The temporal trend of funding and the influence of federal government changes on funding were analyzed using Prais-Winster generalized linear regression. From 2004 to 2020, 1,158 studies were financed (purchasing power parity (PPP$) 230.9 million), with most funding aimed at biomedical research (81.6%) and topics involving dengue, leishmaniasis and tuberculosis (60.2%). Funding was stationary (annual percent change of -5.7%; 95%CI -54.0 to 45.0) and influenced by changes to the federal government. Research funding was lacking for chikungunya, Chagas disease, schistosomiasis, malaria and taeniasis/cysticercosis, diseases with a high prevalence, burden or mortality rates in Brazil. Although the Ministry of Health had several budgetary partners, it was the main funder, with 69.8% of investments. The study revealed that research funding for neglected tropical diseases has stagnated over the years and that diseases with a high prevalence, burden and mortality rate receive little funding. These findings demonstrate the need to strengthen the health research system by providing sustainable funding for research on neglected tropical diseases that is consistent with the population's health needs.

## Author summary

Neglected tropical diseases that mainly affect low-income populations living in developing tropical regions. Examples include dengue fever, malaria and Chagas disease. Brazil is currently one of the countries most affected by these diseases. Research efforts seek solutions to control these diseases around the world, but resources used for this purpose are scarce.

**Funding:** The author(s) received no specific funding for this work.

**Competing interests:** I have read the journal's policy and the authors of this manuscript have the following competing interests:the author G.B.T.M. hold technical position at Ministry of Health of Brazil. The authors A.A.T., E.N.S., L.Y.T.U. and M.T.O. held techinical positions at Ministry of Health of Brazil in differents periods.

In addition, it is still necessary to further investigate the behavior of these diseases in the population, as well as develop new treatments and forms of diagnosis. The present study investigated research funding for neglected tropical diseases by the Ministry of Health between 2004 and 2020, demonstrating its evolution during this period, the diseases most studied and the relationship between the disease, its health situation and research funding in Brazil. This information is essential to diagnose the status of research funding for neglected tropical diseases in Brazil, indicating the need to increase or not the financial resources and which diseases require further research.

## Introduction

Neglected tropical diseases (NTDs) affect more than 1 billion people worldwide, mainly from poor populations living in Africa, Asia and the Americas. The occurrence of these diseases reflects a situation of social vulnerability, which significantly impacts public health and inhibits socioeconomic development [1–3]. Brazil has the largest population in the western hemisphere affected by NTDs and is the most responsible for the burden of these diseases in Latin America, especially in relation to Chagas disease, leishmaniasis, cysticercosis and dengue [4].

The fight against NTDs has become a worldwide concern and is part of a global pact aimed at their eradication through the Sustainable Development Goals (SDGs) and actions of the World Health Organization (WHO) [5,6]. However, there is still much to be done to make this a priority in government policy-making decisions, requiring engagement and political will, as well as sustainable investment in Health Science, Technology and Innovation (ST&I/H) and its alignment with health needs and public health policies [2,7–9].

Investment in ST&I/H enables the advancement of knowledge and subsidizes the implementation, monitoring and evaluation of health policies. However, with respect to NTDs, there is a lack of coordination between health needs and research and development (R&D) funding, and innovative and effective strategies for prevention, diagnosis and treatment that are safe and cost-effective, accessible and adapted to different epidemiological realities must continue to be sought [9–12].

It is noteworthy that the lack of reliable information on R&D funding for NTDs is a worldwide issue [13]. In this respect, due to the sanitary and socioeconomic impact caused by NTDs and the importance of R&D to better understand these diseases and seek solutions to combat them, it is essential to know the current status of R&D funding for this area. This information is necessary to help direct research priorities and resource allocation.

In Brazil, under the auspices of the Federal Government, the Ministry of Health's (MoH) Department of Science and Technology (DECIT in Portuguese) is one of the main funders of health research considered strategic in improving the National Health System (SUS) [14]. The goal of the department is to coordinate, articulate and induce health research, including on NTDs, within the scope of the National Science and Technology System [15,16].

Other studies have described NTD research funding in Brazil, but have only analyzed research funded by specific requests for research on NTDs and not investment evolution over the years [12,17,18]. As such, the present study assessed the evolution of NTD research funding from the Ministry of Health and its partners in relation to investment continuity, diseases studied and the burden of these diseases in Brazil, from 2004 to 2020.

## Method

### Ethics statement

The present study was approved by the Research Ethics Committee of the Faculty of Ceilândia, University of Brasilia, under Certificate of Submission for Ethical Review (CAEE) No. 46003821.0.0000.8093.

### Study context

In Brazil, the National Science, Technology and Health Innovation System includes state and federal agencies and institutions that finance, monitor and evaluate research either individually or collectively (Fig 1A) [19]. In the present study, we analyzed funding for NTD research from the Ministry of Health's Department of Science and Technology (MoH/DECIT) and its partners.

In the federal sphere, within the Ministry of Health, DECIT is responsible for implementing National Policy on Science, Technology and Innovation in Health and coordinating health R&D via intersectoral partnerships for funding health research, particularly with: i) other Ministry of Health departments; ii) the Oswaldo Cruz Foundation (FIOCRUZ)–affiliated with the Ministry of Health and one of the main public institutions that conducts health research in Brazil and produces drugs and biopharmaceuticals for the SUS; iii) the Ministry of Science, Technology and Innovations (MCTI)–responsible for formulating and implementing National Policy on Science and Technology; iv) the Coordination for the Improvement of Higher Education Personnel (CAPES)–affiliated with the Ministry of Education (MEC) and aimed at supporting professors and researchers via research grants and scholarships; v) national funding agencies (such as the National Council for Scientific and Technological Development–CNPq, a funding agency affiliated with the MCTI that funds scientific and technological research and innovation and helps train qualified human resources for research), as well as state (Research Support Foundations–FAPs, public institutions that finance scientific research in Brazilian states) and international funding agencies [15,16,20–22]. It is important to note that ST&I/H falls under the SUS, guaranteed by the 1988 Federal Constitution and Organic Health Law [23,24].

Research funding by DECIT can occur in three ways [25]: a) national open calls for research proposals: selection of research projects at the national level, with the collaboration of the CNPq or other funding agencies; b) state open calls/decentralized support: carried out through the research program managed by SUS (PPSUS), in which research projects at the state level are selected in partnership with the FAPs, CNPq, State Departments of Health and State Departments of Science and Technology, considering the particularities of each Brazilian state; c) Direct Hiring: aimed at addressing the strategic demands of the Ministry of Health or emergency situations in public health, whereby researchers are hired directly without going through a public selection process.

In order to establish research funding priorities in public calls for tender or direct hiring, DECIT uses the National Agenda of Health Research Priorities as a guide, which is based on SUS principles and the national and regional health needs of Brazilians [26]. Appraisal seminars are held to monitor and assess research supported by DECIT (Fig 1B).

### Study design

This is a retrospective, exploratory study with a quantitative approach, aimed at analyzing the evolution of NTD research funding by the DECIT of the Ministry of Health and its partners. Secondary data available in the Health Research repository maintained by DECIT at the

**A – Main public funders of health research in Brazil**

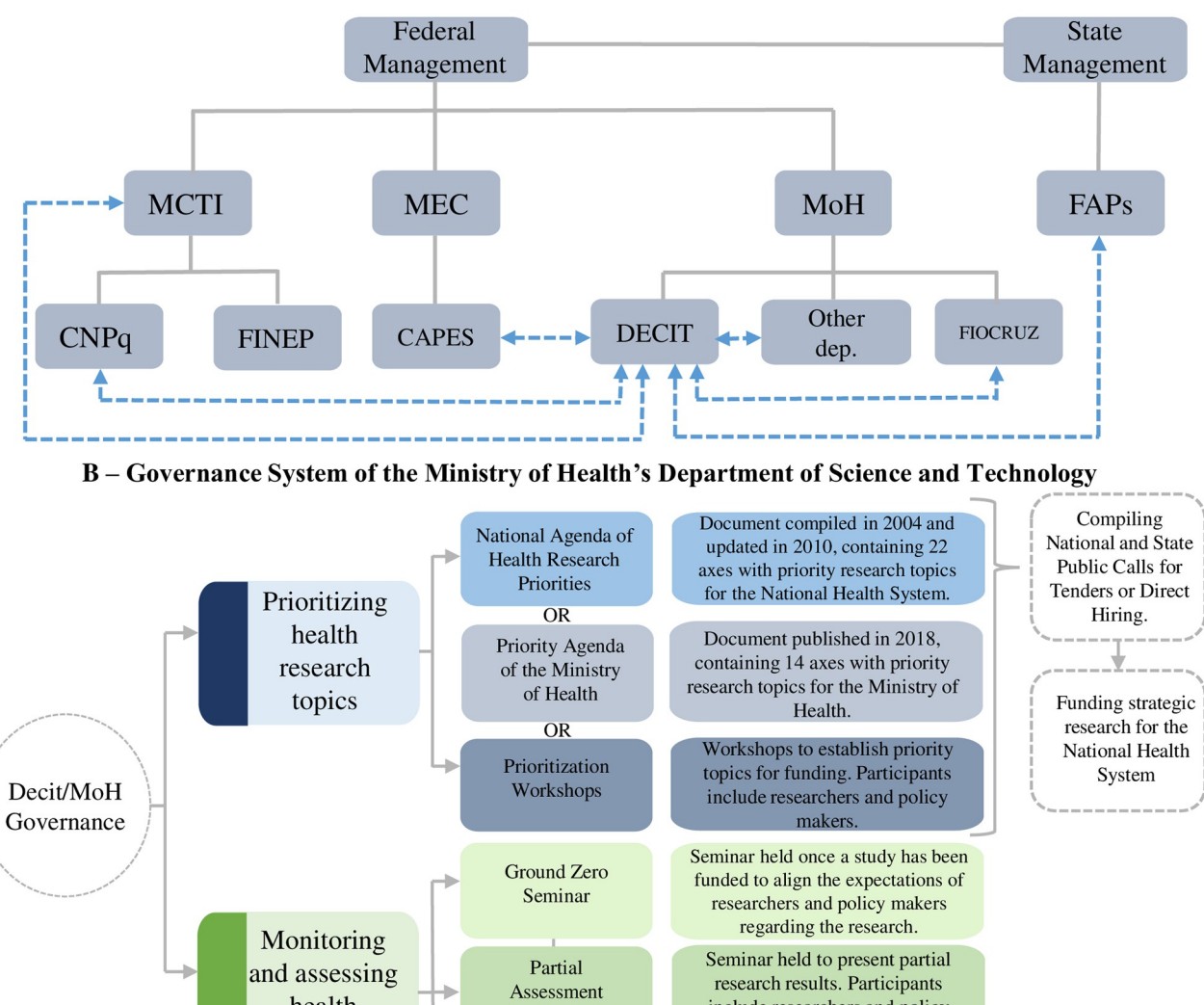

**B – Governance System of the Ministry of Health's Department of Science and Technology**

**Fig 1. Public research funding in Brazil. (A) Main funders of public research. (B) Governance System of health research funding in the Ministry of Health's Department of Science and Technology.** MCTI–Ministry of Science, Technology and Innovations; MEC–Ministry of Education; MoH–Ministry of Health; FAPs–Research Support Foundations; CNPq–National Council for Scientific and Technological Development; FINEP—Funding Authority for Studies and Projects; CAPES–Coordination for the Improvement of Higher Education Personnel; DECIT–Department of Science and Technology; Other dep.–Other Ministry of Health departments; FIOCRUZ–Oswaldo Cruz Foundation. Note: The blue arrows indicate partnerships with the Ministry of Health's Department of Science and Technology, the object of this study, to fund and conduct strategic health research for the National Health System (SUS). Source: Compiled by the authors based on the following publications: National Science, Technology and Innovation Strategy [19]– 2016–2022; National Agenda of Health Research Priorities [26]; Research Priority Agenda of the Ministry of Health [27].

Ministry of Health were used. The time frame considered in the study was between 2004 and 2020, since it comprises the beginning of the Science and Technology Department management of DECIT based on the National Agenda of Health Research Priorities [26] and the availability of data in the Health Research repository.

## Variables and data sources

The search for research data was carried out in June 2022 using as reference 14 endemic NTDs in Brazil, listed by WHO, and included in the SDGs and Global Burden of Disease [1,6,28] study, namely: dengue fever, chikungunya, Chagas disease, echinococcosis, schistosomiasis, filariasis, soil-transmitted helminthiases, leprosy, malaria, onchocerciasis, rabies, taeniasis/cysticercosis, trachoma and tuberculosis. Data were extracted from the Health Research repository [29], which includes research funded by DECIT and its partners.

The collected data were recorded and organized in a Microsoft Excel spreadsheet and the titles and abstracts of the studies were read and analyzed; those related to diseases analyzed in this study were considered eligible and duplicate studies were excluded (40 studies). Eligible studies were stratified by a) year of funding and federal government management; b) type of NTD; c) amount funded; d) source of funding: i) Ministry of Health; ii) MEC/CAPES; iii) MCTI; iv) FAPs and v) National Science and Technology Fund and; d) type of research, according to Canadian Academy of Health Science criteria [30], including: i) biomedical research, which investigates mechanisms of health and disease and produces knowledge about the development of diagnostic, treatment and prevention methods; ii) clinical research, which involves human patients with the aim of improving the diagnosis and treatment of diseases or health conditions; iii) research on health services, which evaluates the health system or services in relation to the organization, funding, access and costs of health care; and iv) population and public health research, which investigates the health determinants of a population. The selection and stratification of NTD research was carried out in pairs and disagreements were resolved by consensus.

## Data analysis

The database was transferred to statistical software R, version 4.1.3, in which statistical analyses were performed and the significance level adopted was 5%. The variables were analyzed using absolute and relative frequency and the main results were exhibited descriptively in tables and figures.

Prais-Winsten generalized linear regression was used to analyze the temporal trend of funding, with a 5% significance level. Beta 1 coefficients ($\beta1$) were estimated, with correction of the first-order temporal autocorrelation and respective confidence intervals of 95%, considering the year in which funding occurred as the dependent variable. The variable "federal government management" was also used, since management changes in the Brazilian federal government can influence research funding. In Brazil, federal government elections are held every four years and generally result in different policy and decision makers in federal organizations such as the Ministry of Health. These changes affect health research funding policies because political decisions by new Ministry of Health managers establish different priorities.

Thus, the different federal government managements were categorized as follows: i) Management 1–2004 to 2006; ii) Management 2–2007 to 2010; iii) Management 3–2011 to 2014; iv) Management 4–2015 to 2016; v) Management 5–2016 to 2018; vi) Management 6–2019 to 2020. Management 1 covered a three-year period that began in 2003 and therefore did not coincide with the start of data collection for our study (2004). Management 4 lasted two years because the president was impeached in 2016, when the vice-president took office for 2 years (2016 to 2018 –Management 5), and Management 6 covered two years, corresponding to the availability of data for this study (until 2020).

The estimated coefficients were used to calculate trends via the Annual Percent Change (APC) parameter and respective 95% confidence intervals (CI), calculated based on the

following formula:

$$APC = [-1 + 10b1 \ x \ 100; CI95\% = [-1 + 10b1min]x \ 100; 1 + 10b1\text{max}] \ x \ 100$$

A positive APC indicates an increasing trend for the time series analyzed, a negative value a decreasing trend, and no statistically significant difference is classified as stationary [31].

The research funding amounts were adjusted by the Broad Consumer Price Index (IPCA) of the Brazilian Institute of Geography and Statistics, based on December 2021 prices, in order to update them to current inflation-adjusted values, and then converted to Purchasing Power Parity (PPP$), with 2021 as the reference year (1 USD = 2.530 BRL), a metric established by the World Bank for international comparisons, which is an alternative to the dollar exchange rate [32].

## Results

Between 2004 and 2020, 1,158 NTD studies were funded, with an investment of PPP$ 230.9 million. The number of studies funded and the amount invested varied over the years, 2006 being the year with the highest number of studies conducted (167; 14.4%) (Fig 2A) and 2008 the highest investment rate (PPP$ 41.7 million; 18.1%) (Fig 2B). There was an increase in the number of funded studies, mainly between 2019 and 2020 (from 7 to 57 studies funded) (Fig 2A).

Analysis of the research funding trend during the study period showed a stationary trend, with an APC of -5.7% (95%CI -54.0 to 45.0). The amount invested in NTD research fluctuated during the period studied, declining sharply from 2006 to 2007, 2010 to 2011, 2014 to 2015 and 2016 to 2017, when changes occurred in the Brazilian federal government. These data indicate that changes in management influenced funding during the study period (p value < 0.05) (Fig 2B).

Research on dengue, leishmaniasis and tuberculosis received the largest investments (PPP$ 64.8 million, PPP$ 59.0 million and PPP$ 38.0 million, respectively). These three diseases together accounted for 56.1% of the number of studies conducted and 60.2% of the resources invested (Fig 3).

Other diseases such as onchocerciasis, trachoma and echinococcosis received little funding, accounting for 0.3% of total NTD research (4 studies) and 0.1% of invested resources (PPP$ 0.3 million) (Fig 3). It is noteworthy that the average amount invested in research for these diseases (PPP$ 0.2 million) was 3 times lower than that for leishmaniasis, dengue and tuberculosis (PPP$ 0.7 million).

When comparing the prevalence (per 100,000 inhabitants), mortality (per 100,000 inhabitants) and burden (disability-adjusted life years–DALY per 100,000 inhabitants) of NTDs with funding from the Ministry of Health and its partners, it was found that for most diseases the investment allocated to research was in accordance with the relevant health needs. However, for chikungunya, Chagas disease, schistosomiasis, malaria and taeniasis/cysticercosis the situation was different, with an imbalance between research funding versus prevalence, mortality or disease burden. Chikungunya and malaria exhibited the 2nd and 3rd highest prevalence among NTDs, but occupied the 9th and 5th position in terms of Ministry of Health funding. Chagas disease, taeniasis/cysticercosis and schistosomiasis showed a high burden (80.4; 38.3 and 31.6 DALY/100,000 inhabitants, respectively) and Brazil occupied the 1st, 2nd and 3rd position compared to the rest of the world in relation to these diseases, respectively; however, in the ranking of research funding by the Ministry of Health it occupied the 4th, 10th and 7th position (Table 1).

With respect to funding by type of research, 81.6% of investments focused on biomedical research, followed by clinical research (11.3%). Research on health services and population and public health received the lowest funding (2.1 and 5.0%, respectively) (Table 2).

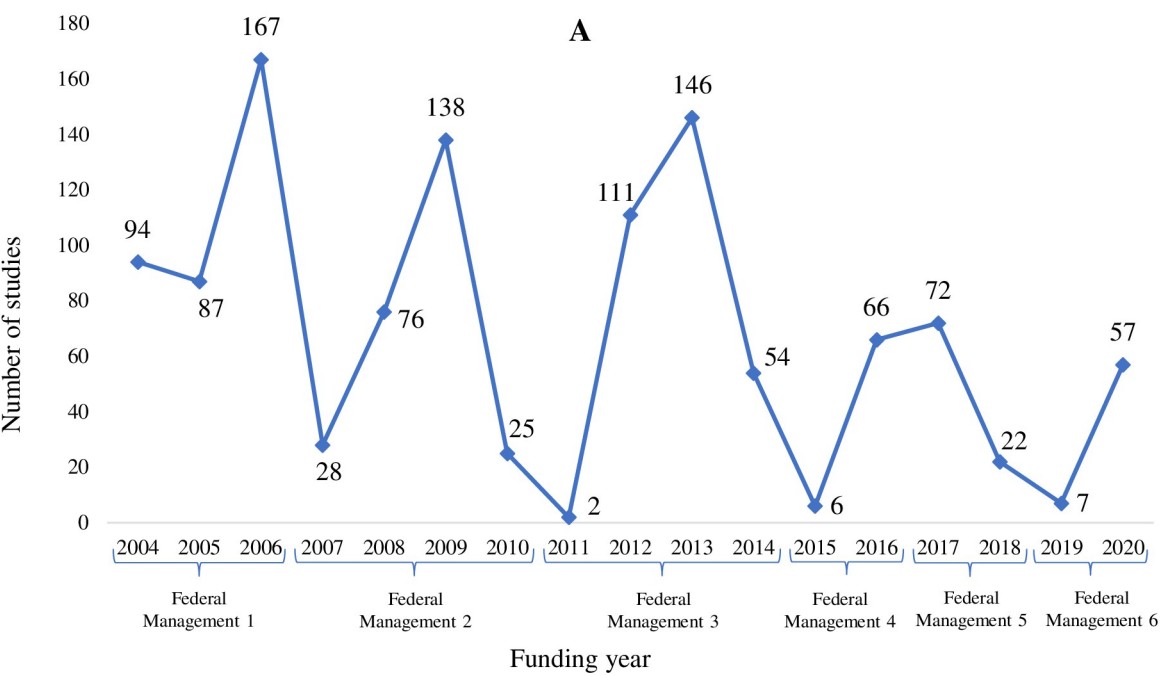

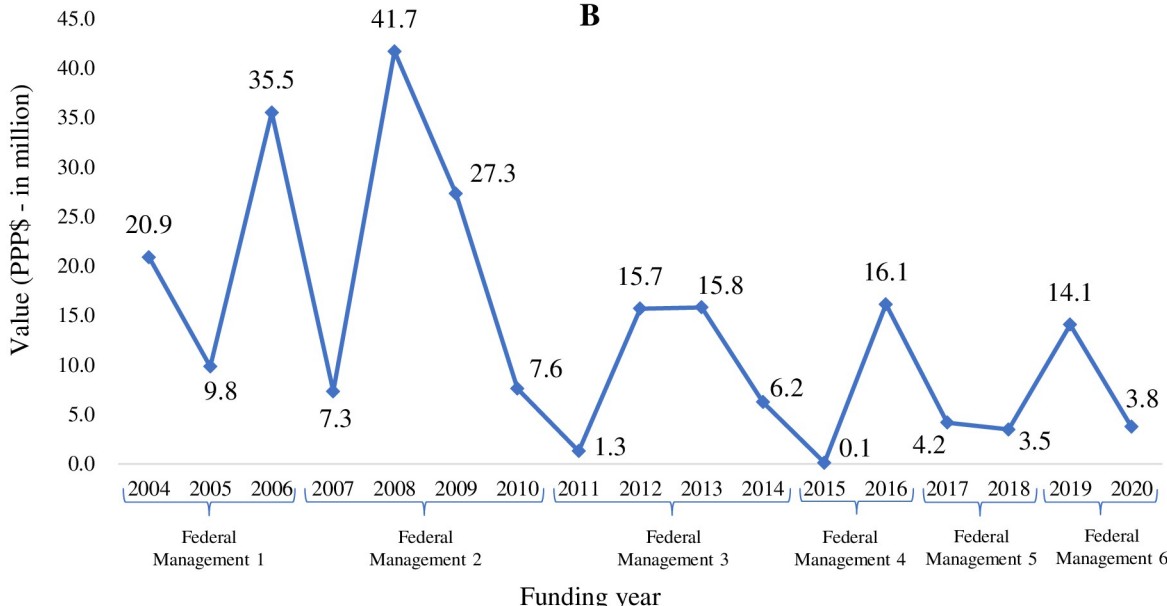

**Fig 2. Distribution of surveys on neglected tropical diseases by year. (A) By number of studies. (B) By amount funded (PPP$). Brazil, 2004 to 2020.** PPP$—Purchasing Power Parity–reference year 2021 (value in million). Source: based on data obtained from the Health Research repository [29], accessed in June 2022.

To fund NTD research, the Ministry of Health received financial support from several partners, as shown in Fig 4(A) and 4(B). Approximately 70% of studies on this topic were funded by resources from the Ministry of Health (PPP$ 161.1 million); however, it is worth noting that research support foundations funded 726 (62.7%) of the 1,158 NTDs studies through the PPSUS program, representing 11.5% of the budget (Fig 4A).

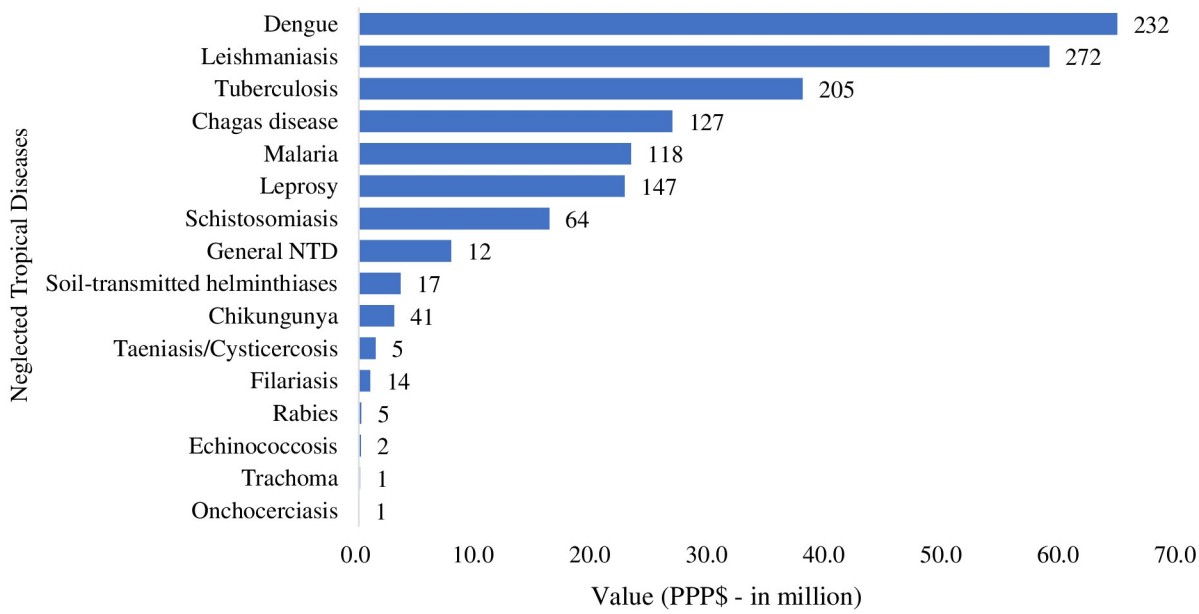

**Fig 3. Distribution of research on neglected tropical diseases and the amount invested for each disease. Brazil, 2004 to 2020.** PPP$—Purchasing Power Parity–reference year 2021 (amount in million); General NTD–Research that encompassed neglected tropical diseases in general, without specifying the disease. Note 1: The number at the top of the bars corresponds to the total number of studies on neglected tropical disease by type of disease. Note 2: 105 surveys encompassed more than one neglected tropical disease and were therefore counted more than once. Thus, the number (1,263 studies) and the amount financed (PPP$ 268.6 million) stratified by type of disease is greater than the number of studies (1,158) and amount funded (PPP$ 230.9 million). Source: based on data obtained from the Health Research repository [29], accessed in June 2022.

Analysis of funding between partners by type of disease showed that the Ministry of Health and research support foundations funded research related to all NTDs. Dengue research received funding from all the partners (Fig 4B).

## Discussion

Between 2004 and 2020, the Ministry of Health, through DECIT and its partners, invested PPP $ 230.9 million in NTD research, totaling 1,158 studies on the subject. Although public funding for research on these diseases showed a stationary trend during this period, changes in the Brazilian federal government influenced the amount invested. The most widely studied and funded research topics were dengue fever, leishmaniasis and tuberculosis, diseases that, despite low mortality rates, exhibit a high prevalence and burden in the country. However, a research funding imbalance was seen in relation to chikungunya, Chagas disease, schistosomiasis, malaria and taeniasis/cysticercosis, even though they are responsible for a high prevalence and burden in the country [4]. Biomedical research absorbed most of NTD funding in the period studied (83.3%), followed by clinical research (10.2%), population and public health research (4.6%) and research on health services (1.9%). In addition, the Ministry of Health was a key entity in funding NTD research, since it contributed 69.8% of the investments and coordinated with several partners at the state, national and international levels in an intersectoral effort to expand investment in research on this topic.

According to G-Finder, a study carried out at United Kingdom (UK) universities found that investments in NTD R&D have increased over the years [38,39]. On the other hand, funding by the Ministry of Health and its partners remained stable, but in Brazil the R&D sector has been experiencing budget cuts, especially after policy changes in the Federal Government

**Table 1. Comparison of prevalence, mortality rate, years of life lost and amount funded by Ministry of Health and partners for research on neglected tropical diseases.** Brazil, 2019.

| Diseases | Region with the highest prevalence of cases in Brazil[1] | Prevalence of confirmed cases per year/per 100,000 inhabitants (2019)[1] | Mortality/per 100,000 inhabitants (2019) | DALY/per 100,000 inhabitants (2019)[3] | Brazil's position in terms of disease burden compared to the rest of the world[3] | Research funding ranking by Ministry of Health and partners (2004–2020)[4] |
|---|---|---|---|---|---|---|
| Chikungunya | Southeast | 84.6 | 0.0 | - | - | 9 |
| Dengue | Southeast | 737.7 | 0.2 | 18.7 | 6 | 1 |
| Chagas disease | North | 0.2 | 2.0 | 80.4 | 1 | 4 |
| Echinococcosis* | - | - | 0.0 | 0.2 | 11 | 13 |
| Schistosomiasis | Northeast | 9.5 | 0.2 | 31.6 | 3 | 7 |
| Filariasis | Northeast | - | 0.0 | 3.2 | 9 | 11 |
| Soil-transmitted helminthiases** | Northeast | 12.6 | 0.0 | 11.0 | 7 | 8 |
| Leprosy | Midwest | 17.4 | 0.1 | 1.2 | 10 | 6 |
| Leishmaniasis | North | 9.4 | 0.1 | 31.2 | 4 | 2 |
| Malaria | North | 74.7 | 0.1 | 5.1 | 8 | 5 |
| Onchocerciasis | North | 0.0 | 0.0 | 0.0 | - | 15 |
| Rabies | South | 0.0 | 0.0 | 0.0 | 15 | 12 |
| Taeniasis/ Cysticercosis* | - | - | 0.0 | 38.3 | 2 | 10 |
| Trachoma*** | North | - | 0.0 | 0.1 | 13 | 14 |
| Tuberculosis | Southeast | 45.5 | 2.1 | 98.3 | 3 | 3 |

DALY–Disability-adjusted life years;

*Diseases that do not have a registration system for cases;

**Data related to the 2010–2015 period;

***Data in the qualification phase, currently not available in the Notifiable Diseases Information System (SINAN) [33]; Green Cells: Research funding by the Department of Science and Technology and partners is aligned with the high prevalence or mortality or DALY of the disease; Orange cells: Research funding by the Department of Science and Technology and partners is not aligned with the high prevalence or mortality or DALY of the disease.

Source:

[1]Notifiable Diseases Information System (SINAN) [33–35];

[2]Mortality Information System (SIM) [36];

[3]Global Burden of Diseases [4];

**National Survey on the Prevalence of Schistosomiasis mansoni and soil-transmitted helminthiases [37]; Based on data obtained from the Health Research repository [29], accessed in June 2022.

**Table 2. Distribution of research funding on neglected tropical diseases by type of research.** Brazil, 2004–2020.

| Research type | Number of studies | | Amount financed | |
|---|---|---|---|---|
| | n | % | PPP$—in million | % |
| Biomedical Research | 805 | 69.5 | 188.5 | 81.6 |
| Clinical research | 205 | 17.7 | 26.2 | 11.3 |
| Population and public health | 76 | 6.6 | 11.3 | 5.0 |
| Health services | 72 | 6.2 | 5.0 | 2.1 |
| **Total** | **1,158** | **100.0** | **230.9** | **100.0** |

PPP$—Purchasing Power Parity–reference year 2021 (value in million).

Source: based on data obtained from the Health Research repository [29], accessed in June 2022.

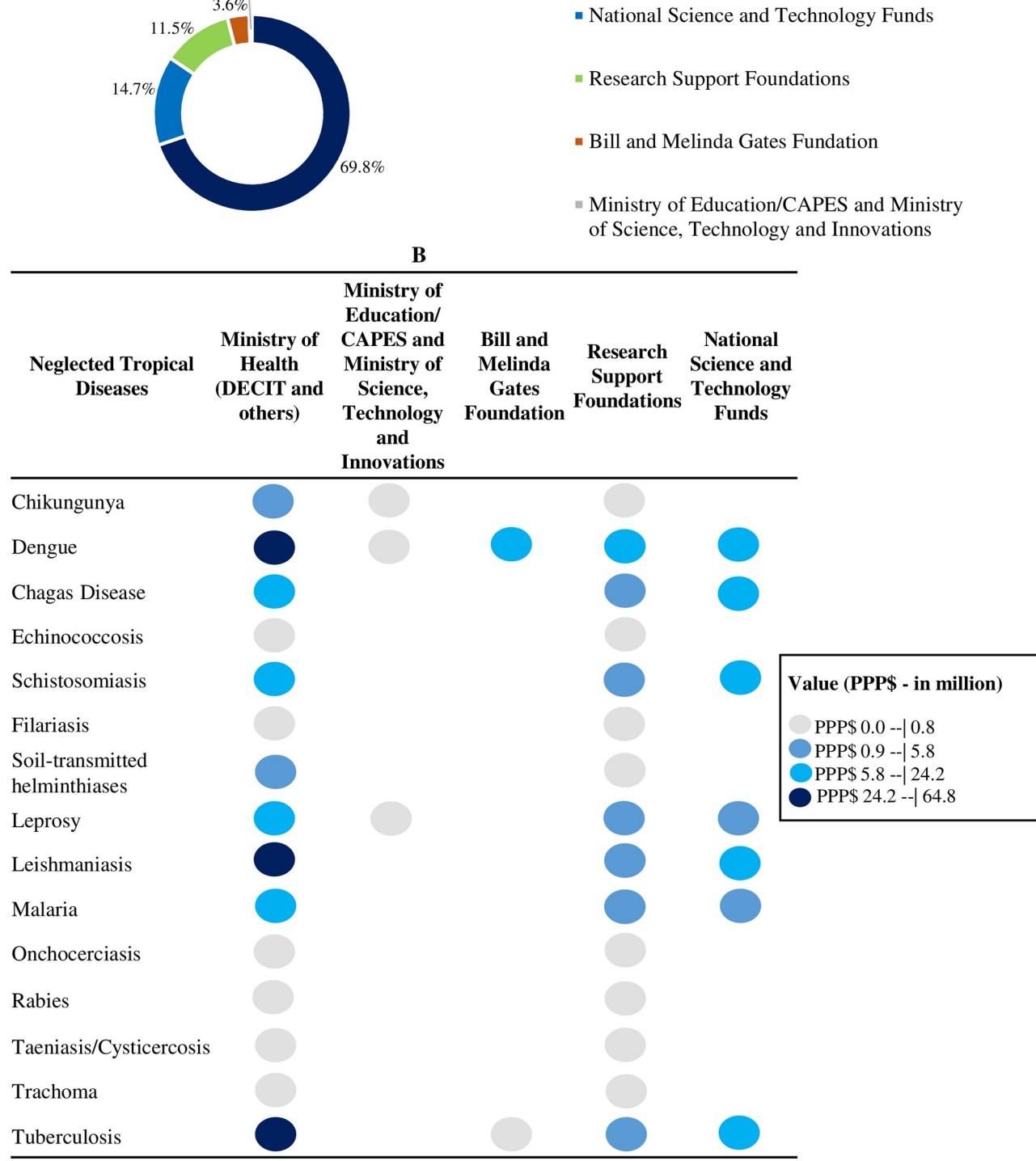

**Fig 4. Distribution of research funding on neglected tropical diseases by the Ministry of Health and partners. (A) Percentage of funding by the Ministry of Health and its partners. (B) Amount funded (PPP$) by the Ministry of Health and its partners by type of disease. Brazil, 2004–2020.** CAPES–Coordination for the Improvement of Higher Education Personnel; DECIT–Department of Science and Technology; MEC–Ministry of Education; PPP$—Purchasing Power Parity–reference year 2021 (value in million). Note: 105 surveys encompassed more than one neglected tropical disease and were therefore counted more than once. Thus, the funded research value stratified by type of disease is greater than the funded amount. Source: based on data obtained from the Health Research repository [29], accessed in June 2022.

in 2016, which has led the country to distance itself from a funding system that guarantees continuity of research and scientific production [40,41].

Therefore, in the future, these cuts may decrease NTD research financing, as occurred in 2020, when Brazil recorded the largest drop in R&D funding for NTDs since 2007 (-57%), especially due to the reduction in funding for DECIT. This decline also occurred in other countries, and may be explained by efforts to combat the Covid-19 pandemic, which gives rise to concern, as it is not known how long the pandemic will influence research funding for other topics [42].

Nevertheless, Brazil ranked 3rd in NTD R&D funders in 2018 among low- and middle-income countries, with an investment of US$ 12 million. However, the amount invested is still small when compared to that spent in the same year by the United States (US$ 1.779 billion), the country that invests the most in NTD R&D in the world, with the North American investment 148 times greater than that of Brazil [43].

In addition to budget cuts, management changes in the Brazilian federal government affected investment in NTD research. It is important to underscore that although the same political party retained control of the federal government from 2004 to 2016, funding for NTD research fluctuated. This suggests that even with the same political party in power, changes occur that alter the direction of decision making on policies involving funding research with public resources. Governments play a key role in implementing policies to address NTDs on a national level, including establishing priorities that guide health research funding [2,44].

Dengue fever, leishmaniasis and tuberculosis received the most research funding, similar to the results found by Fonseca, Albuquerque and Zicker [12]. Dengue constantly causes epidemics in Brazil and has been one of the most challenging NTDs in the world, given its rapid geographic growth, combined with the difficulty in controlling its vector, and developing vaccines and rapid, inexpensive differential diagnostic methods for other acute febrile illnesses [45,46]. In 2019, the global burden of dengue was 30.8 DALYs per 100,000 inhabitants, ranking 2nd in relation to other NTDs. In the same year, Brazil was ranked 6th in terms of the global dengue burden (18.74 DALYs per 100,000 inhabitants) [4].

Among other NTDs in Brazil, leishmaniasis was ranked 4th for disease burden in 2019 [4] and its control remains a distant goal due to the involvement of multiple species, vectors and animal reservoirs, the social stigma surrounding the resulting skin lesions (deformities and scars), lack of accurate diagnostic methods and expansion into previously nonendemic areas [47–49]. Its treatment is also challenging because, despite several studies aimed at alternative therapies, existing options are limited by factors such as toxic side effects, causing poor treatment adherence, low efficacy and resistance to available drugs [50,51].

With regard to tuberculosis, Brazil ranks 3rd in the global burden of this disease, with the WHO considering it a priority country for disease control [52]. The Brazilian public health system provides access to free tuberculosis diagnosis and treatment; however, it is still necessary to prioritize intersectoral public policies and the structuring of health services, as well as invest in research that seeks innovative solutions to control the disease, such as vaccines and the incorporation of new technologies in health systems [53].

On the other hand, chikungunya, Chagas disease, malarial schistosomiasis and taeniasis/cysticercosis, which stand out in terms of prevalence and burden in Brazil, have received little research funding. In addition, not registering echinococcosis and taeniasis/cysticercosis cases and underreporting soil-transmitted helminthiases cases may have masked the real burden caused by these diseases in the country.

It is believed that the small number of chikungunya studies funded is due to the date on which the disease was introduced in the country, since the first autochthonous cases were

reported in 2014 [54]. For taeniasis/cysticercosis, the lack of consolidated public policies to tackle the disease may have resulted in its being "forgotten" in research in the field.

For other diseases, the dearth of research may be related to the introduction of other topics defined as priorities in public calls for tenders for study selection, prompting research groups to focus on topics with available funding. Additionally, the occurrence of malaria, for example, which is concentrated in the Brazilian Amazon Forest in Northern and Midwestern states [55], has made the disease a priority for these regions, meaning that often only researchers from these areas choose to study the topic.

Historically, some diseases have become more neglected than others and, consequently, there has been an investment imbalance between R&D and the burdens of these diseases, especially in developing countries where production and scientific development are usually not guided by NTD-related needs [12,56–58]. To overcome this challenge, it is necessary to plan strategies that guarantee sustainable and equitable funding for actions to combat NTDs, such as the creation of specific research programs, the strengthening of national health research systems with a long-term incentive for R&D, and the establishment of national and international partnerships to maximize fundraising and the effective use of these investments.

The present study found that funding was concentrated on biomedical research. These are studies that investigate health and disease mechanisms, with the aim of developing diagnostic methods, new treatments, and alternatives for disease prevention, through basic research [30]. These studies seek solutions to problems caused by NTDs that still need further investigation, such as fully understanding the forms of transmission, improving prevention methods, and developing accessible treatments and diagnosis methods, given that current technologies are insufficient to alleviate the suffering of the affected populations [59–61]. However, other types of research, especially those with national representation [62], help with surveillance, a key factor in controlling and eliminating NTDs. Thus, equitable investment in all types of research would help the country address NTDs, contributing to achieving the goals proposed in the SDGs related to these diseases.

Research funding for NTDs in Brazil is largely carried out with public resources [42]. As seen in the present study, the Ministry of Health was the main funder of NTD research, but the agency received support from several intersectoral partners that are important for implementing National Policy on Science, Technology and Innovation in Health in the country.

PPSUS is an example of a successful intersectoral partnership, implemented by the Ministry of Health, CNPq, and FAPs in collaboration with the State Departments of Health and Science and Technology, that promotes scientific and technological development to meet the regional health particularities in Brazil [63]. The program was responsible for 62.7% of NTD research observed in the present study and has the potential to incorporate results into health services, since it brings together the stakeholders involved in defining research priorities (researchers, professionals and health managers) [64].

Another success story is the international partnership with the Bill and Melinda Gates Foundation to finance NTD research, such as in the Brazilian Tuberculosis research Network (RedeTB) and World Mosquito Program, whose results have advanced the SUS.

RedeTB aims to provide scientific and technological training for the development of technologies and products to control tuberculosis. One of its achievements is the incorporation of the Xpert MTB/Rif test into the SUS, used to diagnose more than 60% of tuberculosis cases in Brazil [65,66].

The World Mosquito Program, present in 12 countries, is an initiative aimed at combatting arboviruses that replaces *Aedes aegypti* mosquitoes with *Wolbachia* mosquitoes, a bacteria capable of reducing the transmission of dengue, zika, chikungunya and yellow fever. Research

in Brazil has achieved positive results, with a reduction in arbovirus infection rates in areas where *Wolbachia* mosquitoes have been released [67,68].

In addition, partnerships with other federal government ministries include the MCTI through CNPq and the MEC through CAPES, as well as the National Science and Technology Funds, which contribute significantly to the development of government science and technology programs in Brazil [69]. Although the MCTI/CNPq and MEC/CAPES have contributed only 0.5% to NTD research funding, they are partner institutions that help the Ministry of Health with the administrative and financial management of research activities.

The National Science and Technology Funds, which were created to raise funds from different sources to expand science and technology in Brazil, seeking a long-term financing policy [70], leveraged R&D investments in the health sector. However, with the R&D scenario in crisis in Brazil, the MCTI/CNPq, MEC/CAPES and National Science and Technology Funds' budgets have been decreasing over the years, which has hampered scientific and technological development and the training of qualified human resources.

The present study was limited to evaluating the evolution of NTD research funding by the DECIT of the Ministry of Health and some of its partners. It was not possible to include research funded by other Ministry of Health departments because they are not registered on the Health Research platform or in other public repositories. A limitation are studies that received funding but were not registered in the Health Research repository, as well as typing errors or incomplete data, thus hindering the categorization of research projects.

It was concluded that while funding for NTD research remained stable, changes in the federal government influenced the amount invested during the time period studied. The lack of increased funding for NTD research may imply a decline in scientific production and the search for new knowledge that offers solutions to control these diseases.

Dengue, leishmaniasis and tuberculosis were the most studied diseases and received more funding, indicating that they are priorities for the federal government and researchers. On the other hand, few resources were available for studies on chikungunya, Chagas disease, schistosomiasis, malaria and taeniasis/cysticercosis, diseases with a high prevalence, burden or mortality rate in Brazil, which suggests that research funding does not reflect the epidemiological importance of these diseases in the country.

The most funding was allocated to biomedical research. Continued funding for biomedical research is important in obtaining answers to persistent problems involving NTDs; however, this finding demonstrates the need to expand the scope of funding to include other types of research, such as epidemiological studies, in healthcare services and clinics in order to assess the effectiveness of the surveillance strategies, treatments and diagnostics adopted and seek new sustainable solutions.

The Ministry of Health played a leading role in funding NTD research, but also collaborated with several budgetary partners, particularly FAPs. These intersectoral partnerships can contribute to maximizing the capture and use of financial resources.

Finally, the results of this study contribute to subsidizing assessments of governance structure in health research, which must be coherent and based on health needs in order to increase the capacity of health systems, services and policies and improve funding in a scenario of finite and limited resources.

## Acknowledgments

The authors thank the Department of Science and Technology of the Ministry of Health for the opportunity to conduct the present study.

## Author Contributions

**Conceptualization:** Gabriela Bardelini Tavares Melo, Antonia Angulo-Tuesta, Everton Nunes da Silva, Marcos Takashi Obara.

**Formal analysis:** Gabriela Bardelini Tavares Melo, Antonia Angulo-Tuesta, Thaís da Silva Santos.

**Investigation:** Gabriela Bardelini Tavares Melo, Liza Yurie Teruya Uchimura.

**Methodology:** Gabriela Bardelini Tavares Melo, Antonia Angulo-Tuesta, Everton Nunes da Silva, Liza Yurie Teruya Uchimura.

**Visualization:** Gabriela Bardelini Tavares Melo, Everton Nunes da Silva, Thaís da Silva Santos.

**Writing – original draft:** Gabriela Bardelini Tavares Melo.

**Writing – review & editing:** Gabriela Bardelini Tavares Melo, Antonia Angulo-Tuesta, Everton Nunes da Silva, Thaís da Silva Santos, Liza Yurie Teruya Uchimura, Marcos Takashi Obara.

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
