## [Decision Letter · Decision Letter 0]

7 Dec 2022

Dear Ms Melo,

Thank you very much for submitting your manuscript "Evolution of research funding for neglected tropical diseases in Brazil, 2004-2020." for consideration at PLOS Neglected Tropical Diseases. As with all papers reviewed by the journal, your manuscript was reviewed by members of the editorial board and by several independent reviewers. In light of the reviews (below this email), we would like to invite the resubmission of a significantly-revised version that takes into account the reviewers' comments. 

Please carefully consider the comments by both reviewers when revising the manuscript.

We cannot make any decision about publication until we have seen the revised manuscript and your response to the reviewers' comments. Your revised manuscript is also likely to be sent to reviewers for further evaluation.

Sincerely,

Peter Steinmann, Ph.D.

Academic Editor

Dileepa Ediriweera

Section Editor

Please carefully consider the comments by both reviewers when revising the manuscript.

Reviewer's Responses to Questions

**Key Review Criteria Required for Acceptance?**

**Methods**

-Are the objectives of the study clearly articulated with a clear testable hypothesis stated?

-Is the study design appropriate to address the stated objectives?

-Is the population clearly described and appropriate for the hypothesis being tested?

-Is the sample size sufficient to ensure adequate power to address the hypothesis being tested?

-Were correct statistical analysis used to support conclusions?

-Are there concerns about ethical or regulatory requirements being met?

Reviewer #1: It is my feeling that the methods selected are soundly robust to address the aimed objective.

Reviewer #2: The The use of the Prais-Winsten generalized line regression was applied in a time series considering the number of events (studies or $$) during the years. However, these events are strongly associated with the government changes. It is clear that each 3 – 4 years has one year with more funding (peak). The regression applied to the annual number is influenced by the fluctuation, as well the years almost without funding. My suggestion is to control the linear regression using splines for the government changes or just sum the total per period until the election of changes of management. At the end it would be possible to see the reduction in percentages.

**Results**

-Does the analysis presented match the analysis plan?

-Are the results clearly and completely presented?

-Are the figures (Tables, Images) of sufficient quality for clarity?

Reviewer #1: Results presented in tables and figures are quite detailed to allow the reader to follow the discussion.

Reviewer #2: See the suggestions for the anaysis.

**Conclusions**

-Are the conclusions supported by the data presented?

-Are the limitations of analysis clearly described?

-Do the authors discuss how these data can be helpful to advance our understanding of the topic under study?

-Is public health relevance addressed?

Reviewer #1: Yes, the conclusions are directly related to the results presented.

Reviewer #2: Some parts of the conclusion do not depend on the results of this study. Should be based on the evolution analysis.

**Editorial and Data Presentation Modifications?**

Reviewer #1: 1. Gradient of blues in Figure 3 makes difficult to understand which organization is responsible for each percentage or disease. Try using different colors.

Reviewer #2: It is important to correct the analysis, this way the results and discussion would be fitted.

**Summary and General Comments**

Reviewer #1: 1. The article addresses a very important issue, i.e., the shortage of funds for research in health in developing countries. This objective is higly relevant and interesting.

2. I believe authors used a robust methodology to proof their hypothesis and results are quite relevant. 

3. They clearly explain the somewhat confuse brazilian governmental system of research funding. However, my concern is that the internantional reader may remain still confused - anyhow, one cannot blame authors for that.

4. Some of their results are most welcome, such as the imbalance of funding priority among top diseases and the information that biomedical research is by far the type of research mostly funded, in opposition to low percentage of clinical studies, which attests the difficulties of promoting clinical research out of the drug industry area.

5. They also discuss this imbalance in terms of priority of funding and mortality/burden - that is, they complain that chikungunya, Chagas disease, malarial schistosomiasis and taeniasis/cysticercosis, which stand out in terms of prevalence and burden in Brazil, have received little research funding. However, an explanation for this may be the lack of submission of projects in these areas.

6. To some extent the article gives an idea that all governmental funding in health research is provided only by DECIT and its partners - that is not true. By large, most of the governmental health research funding is provided by other federal or state agencies (the partners alone!) that have higher budgets than DECIT. I dare to say that DECIT funds only just a very few number out of the overall health research conducted in the country. Perhaps authors should include a brief note to contextualize this fact.

7. I believe that the findings of this article are quite relevant to the resarch field in Brazil, no doubts. A possible constraint is that the issue is too much specific to our reality and may be difficult to foreign readers to understand the details of the organization of governmental funding for research in Brazil. However, in an overall view, the topic may be also of interest to the world community of health research.

Reviewer #2: This is a very interesting study on the Evolution of research funding for neglected tropical diseases in Brazil, 2004-2021. The manuscript in general is original and relevant mainly for the stakeholders and managers of the research agency, government as well researchers. 

The sequence of the presentation is appropriated, but some points need to be reviewed. 

 Abstract: 

- The results – Follow the topic results (reviewed)

- Conclusion – answer the objective – The recommendation is not the conclusion. 

Introduction: 

Line: 94 – 96

In Brazil, the Ministry of Health’s Department of Science and Technology (DECIT) is currently the main funder of strategic research to improve the public health system and the health status of the population. 

However the reference refers to 2010 and before. Maybe better: …is usually the main…

Results:

The use of the Prais-Winsten generalized line regression was applied in a time series considering the number of events (studies or $$) during the years. However, these events are strongly associated with the government changes. It is clear that each 3 – 4 years has one year with more funding (peak). The regression applied to the annual number is influenced by the fluctuation, as well the years almost without funding. My suggestion is to control the linear regression using splines for the government changes or just sum the total per period until the election of changes of management. At the end it would be possible to see the reduction in percentages.

Discussion:

The author could include some points related to prioritized NTDs as Dengue, TB and Leishmaniasis. For instance: international agenda as Bill & Melinda Gates support. The research group organized a network as TB in Brazil, inducing the MoH support. In the case of Dengue – the burden of disease; and for Leishmaniasis the delay in terms of treatment at the same time of the increased risk area of the incidence. 

Conclusion:

Some parts of the conclusion do not depend on the results of this study. Should be based on the evolution analysis.

PLOS authors have the option to publish the peer review history of their article (what does this mean?). If published, this will include your full peer review and any attached files.

Reviewer #1: No

Reviewer #2: No
---

## [Editor Report · Decision Letter 1]

1 Feb 2023

Dear Ms Melo,

We are pleased to inform you that your manuscript 'Evolution of research funding for neglected tropical diseases in Brazil, 2004-2020.' has been provisionally accepted for publication in PLOS Neglected Tropical Diseases.

Best regards,

Peter Steinmann, Ph.D.

Academic Editor

Dileepa Ediriweera

Section Editor

---

## [Editor Report · Acceptance letter]

20 Feb 2023

Dear Ms Melo,

We are delighted to inform you that your manuscript, "Evolution of research funding for neglected tropical diseases in Brazil, 2004-2020.," has been formally accepted for publication in PLOS Neglected Tropical Diseases.

Best regards,

Shaden Kamhawi

co-Editor-in-Chief

Paul Brindley

co-Editor-in-Chief
